# Unification of Recurrent Neural Network Architectures and Quantum Inspired Stable Design

## Abstract

Various architectural advancements in the design of recurrent neural networks (RNN) have been focusing on improving the empirical stability and representability by sacrificing the complexity of the architecture. However, more remains to be done to fully understand the fundamental trade-off between these conflicting requirements. Towards answering this question, we forsake the purely bottom-up approach of data-driven machine learning to understand, instead, the physical origin and dynamical properties of existing RNN architectures. This facilitates designing new RNNs with smaller complexity overhead and provable stability guarantee. First, we define a family of deep recurrent neural networks, $n$-$t$-ORNN, according to the order of nonlinearity $n$ and the range of temporal memory scale $t$ in their underlying dynamics embodied in the form of discretized ordinary differential equations. We show that most of the existing proposals of RNN architectures belong to different orders of $n$-$t$-ORNNs. We then propose a new RNN ansatz, namely the Quantum-inspired Universal computing Neural Network (QUNN), to leverage the reversibility, stability, and universality of quantum computation for stable and universal RNN. QUNN provides a complexity reduction in the number of training parameters from being polynomial in both data and correlation time to only linear in correlation time. Compared to Long Short Term Memory (LSTM), QUNN of the same number of hidden layers facilitates higher nonlinearity and longer memory span with provable stability. Our work opens new directions in designing minimal RNNs based on additional knowledge about the dynamical nature of both the data and different training architectures.

## Introduction

The invention of neural networks opens astounding directions in bottom-up learning of any universal function $f : x_i \rightarrow y_i$ by optimizing neural network parameters according to existing data pairs $\{x_i, y_i\}$ generated from the function with $x \in \mathcal{X}$ input space, $y \in \mathcal{Y}$ output distribution, $i \in \mathbb{Z}$ index of distribution sample instance. It differs from traditional digital logic circuit encoding of a universal function in two major ways: first, the neural network can be made fully differentiable and thus can morph from one function to the other seamlessly to allow gradient based training; second, knowledge of the function or the model to be learned is not required beforehand, but instead the multitude of data that carries the information of the functional relation becomes necessary.

Although such agnostic learning capability is highly desirable in cases when data are more accessible than the underlying underlying functional form of the model, it also comes with two major limitations. On one hand, to compensate the lack of knowledge about the incoming data, neural network incurs unwanted complexity overhead by requiring more hidden layers and nodes per layer for larger dimensional but not necessarily more complex data. On the other hand, training of deep neural networks with additional hidden layers is difficult due to the lack of stability guarantee. The commonly adopted gradient based optimization of training is proven to fail in most cases due to the observation that gradients of the neural network parameters vanish or explode exponentially quickly in the number of hidden layers (Bengio et al., 1994). This renders the overall learning process to become unstable and volatile to both the training methods and format of the incoming data.

To reduce the complexity overhead, one can relax the strictly agnostic nature of modern neural networks by leveraging the time-sequential nature of data. One outstanding example is in the design of recurrent neural network (RNN). It is realized by looping the output of the last layer of the neural network back as input to the first layer together with additional exogenous variables. If we unroll this recurrent structure, the equivalent number of neural network layers is as large as the number of the recurrence performed to the data, which can be arbitrarily deep.

The recurrent structure, however, does not eliminate the issue of instability of deep neural networks (DNN) but rather amplifies possible inherent instability (Bengio et al., 1994; Pascanu et al., 2013). To mitigate the major limitation of RNNs, various architectural redesigns are introduced to improve the stability. Two of the most successful designs, the Long Short-Term Memory (LSTM) (Hochreiter and Schmidhuber, 1997) and Gated Recurrent Units (GRU) (Cho et al., 2014), for example add gated feedback structure to RNN to store and retrieve long and short-term memory. This improves the reversibility of the architecture for boosted stability. Related work by Hochreiter and Schmidhuber (1997); Cho et al. (2014); Koutnik et al. (2014); Arjovsky et al. (2016a) and Jing et al. (2016) improves the controllability over the self-organization of causal dynamics and offers profound insights into how the length of memory span of RNNs affects the efficiency and quality of the training such as more recently proposed Clockwork-RNN (CW-RNN) (Koutnik et al., 2014) and many others (Wermter et al., 1999; Jaeger et al., 2007; Bengio et al., 2013). Nonetheless, the stability of most RNN architectures is limited to empirical evidence (Cho et al., 2014; Jozefowicz et al., 2015; Karpathy et al., 2015; Alpay et al., 2016; Greff et al., 2017). Moreover, the improved performances are accompanied by the ever increasing complexity of the architecture.

What is the fundamental tradeoff between the stability and complexity of RNN design? Indeed, a large volume of studies have identified the inherent susceptibility of these architectures to instability, and the respective ramifications of such instabilities upon learning by harnessing possible structure of the underlying dynamics of RNN (Hochreiter et al., 2001; Pascanu et al., 2013; Schmidhuber, 2015). In the majority of studies, steady state solution Lyapunov stability analysis was considered, under structural conditions upon the evolution operator, activation function boundness, or bounded time delays settings (Jin et al., 1994; Arik, 2000; Mandic et al., 2001; Cao and Wang, 2003; Zhang and Zeng, 2018). The dynamical system metaphor was often included in the analysis of the RNN as a whole, yet, finer grained analysis of recurring neural network themselves, on a layer by layer basis, using artificial time stepping within the deep neural network itself, was only analyzed (for low-order integration schemes) recently, in the context of deep neural networks (Haber and Ruthotto, 2017). The proposal of unitary evolution RNN (URNN) (Arjovsky et al., 2016b) utilizes a unitary weight matrix to preserve the state variable norm and thus the accumulated gradient. However, the total degrees of the freedom in a unitary matrix scales quadratically as the input data dimension, and incur large memory overhead. To reduce such undesirable complexity, they resort to special ansatz which could otherwise impair the overall performance and representability. In Kosmatopoulos et al. (1995), high order recurrent neural networks have been proposed utilizing high-order products of the network input components from previous states. The framework, only accounted for shallow, non-compositional integration structure, and therefore did not address the possibility of high order integration over artificial (layers) time steps. The significance of depth and compositional structures has been provably realized in recent work (Mhaskar and Poggio, 2016).

If RNN is used to learn temporal structures from data produced in the physical world, why not seek inspirations from physics for RNN designs? One counterpart of a neural network in a physical system is outlined by Richard Feynman, who proposed the first use of a quantum computer to conduct any classical computations reversibly based on the laws of quantum mechanics described by linear algebra and linear ordinary differential equations (ODE) in (Feynman, 1986). Feynman provides an exact map between any universal function and the dynamics of a quantum mechanical system. Such a map, similar to that of a neural network, is also fully differentiable in its parameters. Moreover, it also linear and reversible due to the fundamental linear nature of quantum mechanics. Ever since Feynman's historical paper, continuous efforts have resulted in surprising discoveries in quantum physics and a plethora of quantum algorithms, e.g. (Shor, 1994; Grover, 1996), that possess provable advantages over the classical counterparts. A crucial connection remains to be made, between quantum dynamics and the dynamics of propagation in a neural network before one can utilize fruitful results in the quantum realm to design better RNNs.

RNN formulation has known several transformations, which facilitated deeper insights into regarding both the dynamical process of RNN and the underlying physics behind the data. The first thrust of

|          | LSTM      | GRU       | URNN      | CW-RNN    | QUNN      |
|----------|-----------|-----------|-----------|-----------|-----------|
| quantum  | 2-$L$-ORNN | 2-$L$-ORNN | 2-$L$-ORNN | $L$-$L$-ORNN | $L$-$L$-ORNN |

Table 1: Categorization of some of existing RNN architectures according to its physical ODE counterpart. LSTM (Hochreiter and Schmidhuber, 1997): long-term short-term memory RNN with $L$ hidden layers. GRU (Cho et al., 2014): gate model recurrent neural network with $L$ hidden layers. URNN (Arjovsky et al., 2016a): unitary evolution recurrent neural network with $L$ hidden layers. CW-RNN (Koutnik et al., 2014): clockwork recurrent neural network with $L$ hidden layers. QUNN: quantum universal computing recurrent neural network with $L$ hidden layers. $n$-$t$-ORNN: recurrent neural network that corresponds to a discrete integration of ode equation using $n^{th}$ order integration method and up to $t^{th}$ order gradient.

RNN revolution is based on our knowledge of the time sequentiality of the data, which addresses the complexity overhead of conventional deep neural network, but worsened the stability problem. The second wave of RNN proposals addresses the stability by ad hoc architectural changes that use only heuristic and empirical knowledge of the underlying mechanics. This leads us to attempt at establishing theoretical foundations for RNN and neural network in general, initiated by recent works such as Arjovsky et al. (2016b); Mhaskar and Poggio (2016) and Haber and Ruthotto (2017), to apply physics with numerical stability analysis used in solving physical problems for understanding and designing new machine learning architectures.

Towards establishing this third wave of RNN revolution, we formulate a framework that unifies different proposals of RNN architectures according to the order of nonlinearity and the order of temporal memory scale of the underlying dynamics embodied in the integration of ODEs, see Table. 1. To establishe the exact connection between quantum dynamics and RNN dynamics, we show in Sec. 2 that the update rules of RNN can indeed describe the model proposed by Feynman for encoding universal functions in a reversible manner. Lastly for reducing complexity while providing stability guarantees, in Sec. 3 we propose an embedding, called Quantum inspired Universal computing recurrent Neural Netowrk (QUNN), with a complexity of the optimization that grows linearly with the temporal correlation length between input data but independent of the dimension of data itself.

## 1 STABLE RECURRENT NEURAL NETWORK

The success of supervised machine learning techniques depends on the stability, the representability and the computational overhead associated with the proposed training architecture. A generic neural network without any structure, however, is susceptible to exploding or vanishing gradients and requires additional heuristic optimization techniques to suppress such instability.

The stability of deep neural networks can be understood by a simple connection between the integration of discrete ODE and neural network forward and backward propagation (Haber and Ruthotto, 2017). Take a type of stable deep ResNet proposed by Haber et. al. for example: let $l^{th}$ layer of hidden variable be $Y_l \in \mathcal{R}^{s \times p}$ and bias be $b_l \in \mathcal{R}^{s \times p}$, to ensure the stability of propagation, they introduce a conjugate variable $Z_{l \pm \frac{1}{2}} \in \mathcal{R}^{s \times p}$ as a intermediate step such that the propagation of neural network is described by

$$Z_{l+\frac{1}{2}} = Z_{l-\frac{1}{2}} - h_l \sigma(W_l^T Y_l + b_l), \ Y_{l+1} = Y_l + \sigma(W_l Z_{l+\frac{1}{2}} + b_l). \tag{1}$$

The dynamics of the above discrete ODE is stable regardless of the form of weight matrix $\hat{W}_l$ (Haber and Ruthotto, 2017).

In this section, we will extend upon the approach first introduced by Haber and Ruthotto (2017) to include existing architectures of RNN, such as LSTM, GRU, UNN and CW-RNN, as special instances under the a unified framework. We first define an ODE recurrent neural network with $n^{th}$ order in nonlinearity and $t^{th}$ order in time-derivative ($n$-$t$-ORNN) according to its propagation rule: the update of $n$-$t$-ORNN can be mapped to a generalized $n^{th}$ order Runge–Kutta integration. The specific choice of Runge-Kutta method is not essential to such generalization, and can be replaced by other integration method as well. We then analyze the corresponding $n$-$t$-ORNN family different existing RNN architectures belongs. Lastly, we define the $n$-2-ORNN with anti-Hermitian weight matrices as $n$-ARNN and prove the stability of 1-ARNN and 2-ARNN.

| | traditional RNN (LSTM) | physical RNN (ORNN) |
|---|---|---|
| $Y_l$ | input at time step $l$ | state variable at time step $l$ |
| $K_{l_j}$ | $j^{th}$ hidden layer | $j^{th}$ order increment of the gradient slope |
| $\gamma_{l_j}$ | forget gate activation | energy dissipation rate |
| $\alpha_{i,j}$ | weight matrix for hidden variable | weight of $i^{th}$ increment in $j^{th}$ order slope |
| $\kappa_{l_j}$ | input gate's activation | re-scale factor of normalized gradient function |
| $\sigma_{l_j}$ | activation function of $j^{th}$ hidden layer | gradient function |

Table 2: Comparison of the LSTM architecture and $n^{th}$ order ORNN structure.

## 1.1 $n^{th}$ ORDER ODE RECURRENT NEURAL NETWORK

**Definition 1.** An ODE recurrent neural network of $n^{th}$ order in nonlinearity $t^{th}$ order in gradient ($n$-$t$-ORNN), with integers $n, t \geq 1$ and $k \in [n]$, is described by the update rule between input state value $Y_l \in \mathcal{R}^s$ at time step $l$, the hidden variables of the $k^{th}$ layer as $K_{l_k} \in \mathcal{R}^p$ with $1 \leq k \leq n$, and output state value $Y_{l+1} \in \mathcal{R}^s$ as

$$K_{l_1} = \sigma_{l_1}\left(W_{l_1}Y_l + b_{l_1}\right), \; K_{l_q} = \gamma_{l_q}K_{l_{q-1}} + \kappa_{l_q}\sigma_{l_q}\left(W_{l_q}Y_{l+t_{q-1}} + b_{l_q} + h\sum_{k=1}^{q-1}\alpha_k \circ K_{l_k}\right),$$
(2)

$$Y_{l+t_n} = \gamma_{l_{n+1}}Y_l + \kappa_{l_{n+1}}\sigma_l\left(W_{l_n}Y_{l+t_{n-1}} + b_{l_{n+1}} + h\sum_{k=1}^{n}\beta_k K_{l_k}\right)$$
(3)

where $2 \leq q \leq n$; the time corresponding to each hidden layer obeys $t_k = \lfloor t\frac{k}{n} \rfloor$ with the overall time steps shared by the $n$ hidden layers being $t$; the pointwise activation function $\sigma_*(\circ) : \mathcal{R}^n \to \mathcal{R}^q$ at each layer is a nonlinear map that preserves the dimension of the input $q$; the weight matrix at each layer is represented by $W_{l_*} \in \mathcal{R}^{q_2 \times q_1}$ where $q_1$ is the dimension of the input variable and $q_2$ is the dimension of the output variable; $\beta_k, \gamma_{l_m}, \kappa_{l_m}$ and $\alpha_{jk} \in \mathcal{R}^{p \times p}$ are matrices served to rescale and rotate the hidden variables; $\gamma_k$ is a square matrix that manifests the energy dissipation of the ODE dynamics. Below, we analyze some of the most widely used RNN architectures in regard to the nonlinearity and memory scale of their underlying dynamics.

**Claim 1.** Both LSTM and GRU belong to the 2-$L$-ORNN.

*Proof*: For one layer RNN, we have the update rule for LSTM (Hochreiter and Schmidhuber, 1997) as:
$$K_t = f_t \circ K_{t-1} + i_t \circ \sigma_2(W_c Y_{t-1} + U_c K_{t-1} + b_c), \; Y_t = o_t \circ \sigma_1(K_t)$$
(4)

with vector coefficient determined by

$$f_t = \sigma(W_f Y_{t-1} + U_f K_{t-1} + b_f),$$
(5)
$$i_t = \sigma(W_i Y_{t-1} + U_i K_{t-1} + b_i), \; o_t = \sigma(W_o Y_{t-1} + U_o K_{t-1} + b_o)$$
(6)

which is equivalent to setting $n = 2$, $t = 1$, $\gamma_{l_{m_2}} = D[f_t], \kappa_{m_2} = D[i_t], W_{l_{m_2}} = W_c, b_{l_{m_2}} = b_c, h\alpha_{21} = U_c$ and $\gamma_{l_m} = 0, \kappa_{l_m} = o_t$ in $n$-$t$-ORNN. Notice that the weight matrix in ORNN can depend on time and is therefore able to include the memory dependency from $K_{t-1}$. We use $D[a]$ to represent a $p \times p$ diagonal matrix with each diagonal element equal to each element of the vector $a$ of length $p$. This is because the Hadamard product between two vectors can be re-written as diagonal matrix matrix multiplication with the second vector: $a \circ b = D[a]b$.

For multi-layer LSTM with $L$ hidden layers, the only change is that the diagonal matrices $D[f_t], D[i_t]$ and $D[o_t]$ are generalized to $D[f_t^l], D[i_t^l]$ and $D[o_t^l]$, which not only depend on the hidden variable of the same layer from the previous time step, but also the hidden variable of the same time step from a previous layer:

$$f_t^l = \sigma(W_f K_t^{l-1} + U_f K_{t-1} + b_f^l), \; i_t^l = \sigma(W_i K_t^{l-1} + U_i K_{t-1} + b_i^l), \; o_t^l = \sigma(W_o K_t^{l-1} + U_o K_{t-1} + b_o^l)$$
(7)

where $K_t^0 = Y_{t-1}$, and thus the nonlinearity of the ODE increases by one when the number of hidden layers increase by one, thus gives $L$-2-ORNN for a $L$ layer architecture.

For one layer GRU (Cho et al., 2014), we have the update rule as:

$$Y_t = (1 - z) \circ Y_{t-1} + z \circ \tanh\left(W_t Y_{t-1} + W_g r \circ Y_{t-1}\right), \text{ with } z = \sigma(W_l^z Y_{t-1}), \ r = \sigma(W_l^r Y_{t-1}) \tag{8}$$

we can rewrite $r \circ Y_{t-1}$ as $\sigma'(W_l^q Y_{t-1})$ and thus simplify the update rule to

$$Y_t = (1 - z) \circ Y_{t-1} + z \circ \tanh\left(W_t Y_{t-1} + W_g \sigma'(W_l^q Y_{t-1})\right) \tag{9}$$

which is equivalent to setting $n = 1, \gamma_{l_{m_1}} = D[(1-z)], \kappa_{m_1} = D[z], W_{l_{m_1}} = W_t, b_{l_{m_2}} = 0, h\beta_1 = W_g$ in $n$-ORNN. This can be similarly generalized to multi-layer GRU with $L$ total hidden layers by allowing the weight matrices $D[(1 - z)]$ and $D[z]$ to also depend on same layer hidden variable of previous step:

$$z = \sigma(W_l^z Y_{t-1}^l + W_l^{z'} Y_t^{l-1}), \ r = \sigma(W_l^r Y_{t-1}^l + W_l^{r'} Y_t^{l-1}) \tag{10}$$

which for $l^{th}$ layer it corresponds to $L$-2-ORNN. Q.E.D.

**Claim 2.** Unitary evolution RNN (Arjovsky et al., 2016a) with $L$ hidden layers belongs to the $2$-$L$-ORNN.

*Proof*: The propagation rule of URNN between the input to the RNN at time step $1 \le t \le T$: $Y_t$ and hidden variables at the same time step $K_t$ and output to the RNN of the same time step $t$ as $Y_{t+1}$ is:

$$K_{t+1} = \sigma(W_l K_t + V_l Y_t), \text{ for } 1 \le l \le t, \ Y_{t+1} = W_{l+1} K_{t+1} + b_{l+1}, \tag{11}$$

which corresponds to setting $\gamma_{j_j} = 0$ and choosing $n = 2$ and $t = L$ in Eq. (2)–(3). URNN thus belongs to 2-$L$-ORNN. Q.E.D.

**Claim 3.** Clockwork RNN (Koutnik et al. (2014)) with $L$ clocks belongs to the $L$-$L$-ORNN.

*Proof*: The propagation rule for CW-RNN between input $Y_t$ at time step $t$, hidden layers at the same time step $K_t$ as well as from the previous time step $K_{t-1}$ and output $Y_{t+1}$ is described by:

$$K_t = \sigma_h\left(W_H(t)K_{t-1} + W_I(t)Y_t\right), \ Y_{t+1} = \sigma_o\left(W_o K_t\right) \tag{12}$$

where the time-dependent weight matrices $W_H(t)$ and $W_I(t)$ are structured to store memory of previous time steps in into different blocks with increasing duration of time delays such that effectively one can rewrite $W_H(t)K_{t-1} = \sum_j W_j \sigma_j^{t-j-1} W_H(j)K_j + W_I(j)Y_j$ contributions from all previous step iteratively, and so is the clock structure in $W_I(t)Y_t$ which contributes to all hidden layers after $t$. This is equivalent to setting $n = t = L$ and $\gamma_{j_j} = 0$ and $t = 1$ in Eq. (2)–(3). CW-RNN thus belongs to $L$-$L$-ORNN. Q.E.D.

**Definition 2.** The $n^{th}$ order ODE anti-Hermitian recurrent neural network ($n$-ARNN) corresponds to setting all weight matrices in $n$-2-ORNN with anti-Hermitian matrices.

**Theorem 1.** 1-ARNN with monotonic activation function $\sigma_*(\cdot) : \mathcal{R}^n \to \mathcal{R}^n$ and purely imaginary anti-Hermitian weight matrix is stable for small enough $h$ such that $|h \max_k \lambda[W_{l_k}]| < 1$.

*Proof*: This will be proven in Theorem 4, where the original complex anti-Hermitian matrix is embedded into a Hilbert space twice as large such that a purely imaginary anti-Hermitian weight matrix guarantees the stability of the first order integration method. Q.E.D.

**Theorem 2.** Both 2-ARNN and 1-ARNN are reversible.

*Proof*: Since 2-ARNN corresponds to the first order mid-point integration and 1-ARNN corresponds to the symplectic Euler intergration their reversibility is guaranteed by the reversibility of these two integration schemes inside the stble regime. Q.E.D.

It is notable that the definition of $n$-$t$-ORNN does not restrict weight matrices to be time independent. This setup is less restrictive than conventional definition of RNN and is indispensable for generalizing various architectures of RNN under the same framework. Such generalization, however, is well-founded in ODE framework: a generic ODE does not have to be time-independent.

The unification of different RNN architectures through $n$-$t$-ORNN prompts a better usage and obtainment of the memory scale and degree of nonlinearity of the data to further reduce the necessary complexity in the learning architecture. We demonstrate such combination between bottom-up data centered and top-down model centered approach in finding more efficient RNN design in the next section.

|  | column vector | row vector | matrix | inner product | tensor product | Hadamard product |
|---|---|---|---|---|---|---|
| classical | $Y_l$ | $Y_l^T$ | $W_l$ | $Z_l^T Y_l$ | $Y_l \otimes Z_l$ | $Y_l \circ Z_l$ |
| quantum | $|Y_l\rangle$ | $\langle Y_l|$ | $\hat{W}_l$ | $\langle Z_l|Y_l\rangle$ | $|Y_l\rangle \otimes |Z_l\rangle$ | $\hat{D}[Y_l]|Z_l\rangle$ |

Table 3: Comparison of the representation of linear algebra in quantum and in classical literature.

## 2 QUANTUM DYNAMICS AND PROPAGATION OF RECURRENT NEURAL NETWORK

In this section, we start by defining mathematical notations that make the connection between quantum and classical computation more self-evident. We also review previous work by Feynman in reversible classical computation through quantum dynamical evolution. Since the underlying mathematical structure of quantum mechanics is none other than linear algebra, Table. 3, represents the required mapping between quantum and classical audiences. For the sake of consistency, we will henceforth adopt the quantum mechanical notations of linear algebra.

Subsequently, we will establish a key connection between RNN and quantum dynamics: the equivalence between the discretized evolution of a quantum system and the propagation of an RNN up to inversely polynomial errors in the total time steps and spectrum norm of the ODE characteristic function. Since any existing classical solver is discrete in nature but can solve time-dependent quantum dynamics to a given accuracy, our results is general. Such specific connection forms the basis of our stable and efficient RNN ansatz to be discussed in the following section.

The dynamics of a quantum system can be described by a first order ordinary differential equation, namely the Schrödinger equation, where the quantum state parameter represented by the complex vector $|\psi(t)\rangle$ at time $t$ obeys:

$$\frac{d}{dt}|\psi(t)\rangle = -i\hat{H}|\psi(t)\rangle, \tag{13}$$

where $\hat{H}$ is called the quantum Hamiltonian that determines the dynamical evolution of the state parameters. Stepping back to the world of linear algebra, the Hamiltonian matrix is essentially the gradient with respect to time in the first order linear ODE. Despite such fundamental linearity, the emergent phenomena in a sub-region of the quantum system can be highly nonlinear and intriguing.

The power of quantum dynamics in computation was first demonstrated by Richard Feynman in his proposal of a quantum computer: any Boolean function can be encoded into reversible evolution of a quantum system under a carefully chosen system Hamiltonian with total number of Hamiltonian terms equal the number of logical gates that describe the given Boolean function.

**Theorem 3.** Any uniform family of Boolean functions $f : \{0,1\}^n \to \{0,1\}^n$ can be mapped to a unique fixed point of ODE evolution with its characteristic function containing polynomial in $n$ many parameters.

The complete proof is given in references (Feynman, 1986; Kitaev et al., 2002; Aharonov et al., 2008) and is reviewed in supplementary material A. To utilize this result for RNN design, we establish the connection between RNN propagation and discretized quantum evolution in Theorem 4.

**Theorem 4.** Evolution of a closed quantum system of dimension $2^n$ under the Hamiltonian $\hat{H}(t)$ for time $T$ can be approximated by the stable propagation of an RNN with anti-Hermitian weight matrix of size $2^{n+1}$, using $T||\hat{H}||_\infty$ timestep and incurring errors of order $O\left((1/T||\hat{H}||_\infty)^2\right)$.

*Proof*: It is shown by McKague et al. (2009) that any complex Hamiltonian can be mapped to a real Hamiltonian $\hat{H}(t)$ at any time $t$ with constant overhead in computation basis. Using their results, we assume our quantum Hamiltonian is real in each element, and separate the real part $|P\rangle$ and imaginary part $|C\rangle$ of the quantum state as $|\psi\rangle = |P\rangle + i|C\rangle$. After discretization, the ODE that corresponds to Eq. (13) can be solved by symplectic Euler integration that is represented by an RNN with anti-symmetric weight matrix and identity activation function (Arjovsky et al., 2016b):

$$\begin{bmatrix} P(n+1) \\ C(n+1) \end{bmatrix} = \begin{bmatrix} I_{2^{n+1} \times 2^{n+1}} & \delta t \hat{H}(n) \\ -\delta t \hat{H}(n) & I_{2^{n+1} \times 2^{n+1}} - \delta t^2 \hat{H}^2(n) \end{bmatrix} \begin{bmatrix} P(n) \\ C(n) \end{bmatrix} \tag{14}$$

with stable regime bounded by $|\delta t||\hat{H}||_\infty| < 2$ satisfiable by $|\delta t||\hat{H}||_\infty| < 1$. This gives a minimum time step of $T||\hat{H}||_\infty$ and in turn an error of order $O\left((1/T||\hat{H}||_\infty)^2\right)$. Q.E.D.

## 3 QUANTUM INSPIRED UNIVERSAL COMPUTING NEURAL NETWORK

To design stable RNN architectures without incurring unwanted complexity, we propose a new RNN ansatz that uses input data itself to construct weight matrices: QUNN based on our physical knowledge of quantum computation and the proof of equivalence between RNN and quantum evolution.

To facilitate a more efficient training process, we reduce the total degree of freedom in the training parameters of our quantum inspired RNN ansatz from polynomial in both input data size and time-correlation length, i.e., the memory range, of the dynamics, to only linear in the memory range. We achieve this by constructing time-dependent weight matrices in hidden layers from the input data. Such construction makes the most out of the knowledge of manifested time-correlated structure of the data itself in place of a conventional weight matrix ansatz that is data independent. QUNN is thus able to speedup training processes where input data are exponentially larger than the correlation time of the data structure.

**Definition 3.** Quantum inspired universal computing neural network: a recurrent neural network architecture that adopts the update rule between an input state $|Y_l\rangle$, which could be either a binary, real, or complex vector depending on the problem type, and output of the network $Y_{l+1}$ at the integer time step $l \in \{1, 2, ...., N\}$ according to three stages. In the first emedding stage the incoming data is transformed into itself tensor producted with a clock state that marks the relative time sequence through the embedding weight matrix $\hat{E}_l$:

$$|K_l\rangle = \sigma_1\left(\hat{E}_l|Y_l\rangle\right) = \sigma_1\left(|Y_l\rangle \otimes |l\rangle\right), \quad \hat{E}_l = |l\rangle \otimes \sum_j |j\rangle\langle j| \tag{15}$$

where $\sigma_i(\cdot)$ represents the monotonic and continuously differentiable point-wise nonlinear function of the $i^{th}$ layer. In the second stage, the hidden layer $|K_l\rangle$ is updated according to

$$|K_l'\rangle = \hat{S}_1|K_l\rangle + \sigma_2\left(\hat{H}_l|K_l\rangle\right), \tag{16}$$

$$\hat{H}_l = \hat{D}_l\hat{W}_l \tag{17}$$

$$\hat{D}_l = (1 - p_1(l))I \otimes I^c + p_1(l)\left(I \otimes |l+1\rangle\langle l|^c - I \otimes |l\rangle\langle l+1|^c\right) \tag{18}$$

$$\hat{W}_l = (1 - p_2(l)\hat{W}_{l-1} + p_2(l)\left(|Y_{l+1}\rangle\langle Y_l| \otimes |l+1\rangle\langle l| - |Y_l\rangle\langle Y_{l+1}| \otimes |l\rangle\langle l+1|\right) \tag{19}$$

where the $D_l$ consists of identity matrix weighted by $1 - p_1(l)$ and a time re-ordering operator weighted by $p_1$. Notice that $|1\rangle\langle l-1|^c$ is the generator of the permutation group of clock states which adds noise as well as correction to possibly mislabeled time sequence of training data. Such a dispersion step is in product with $\hat{W}_l$, which records the flexible range of memory important for the training: the dependence of $p_2(l)$ on time step $l$ will determine how long the memory lasts.

In the last stage, the state is mapped back to the original dimension by projecting on to the corresponding clock state of the next time step:

$$|Y_{l+1}\rangle = \hat{S}_2|Y_l\rangle + \sigma_3\left(\hat{U}_l|K_l'\rangle\right) = \sigma_3\left(\sum_j |j\rangle\langle j| \otimes \langle l+1|K_l'\rangle\right). \tag{20}$$

Notice that our weight matrix ansatz depends on the time step $l$, which is kept from the first input until the last input of the same set of time-sequential data and is reset to 1 at the beginning of each time-sequential data set. This is different from conventional definition of RNN where the weight matrix does not explicitly depend on time, but such memory dependence is indirectly actuated through the gate construction such as the forgetting unit $f_i$.

The multi-layer generalization of the weight matrix construction in Eq. (19) to facilitate longer and longer time step rotation as the hidden layer number $l_n$ increases takes the form:

$$\hat{D}_{l_n} = (1 - p_1(l_n))I \otimes I^c + p_1(l_n)\left(I \otimes |l\rangle\langle l-n|^c - I \otimes |l-n\rangle\langle l|^c\right) \tag{21}$$

$$\hat{W}_{l_n} = (1 - p_2(l_n)\hat{W}_{(l-1)_n} + p_2(l_2)\left(|Y_l\rangle\langle Y_{l-n}| \otimes |l+n\rangle\langle l| - |Y_{l-n}\rangle\langle Y_l| \otimes |l\rangle\langle l+n|\right), \tag{22}$$

which can be understood as effectively implementing a different order of integrating a time-dependent Hamiltonian evolution of discretized Eq. (13), as illustrated in Fig. 1.

The existence of QUNN is guaranteed by Theorem 1. Moreover, the Hermitian matrix as the weight matrix of a neural network also secures its stability: its eigenvalues are always purely imaginary and thus guarantee the stability of the QUNN when choosing a leapfrog integration method (Haber and Ruthotto, 2017).

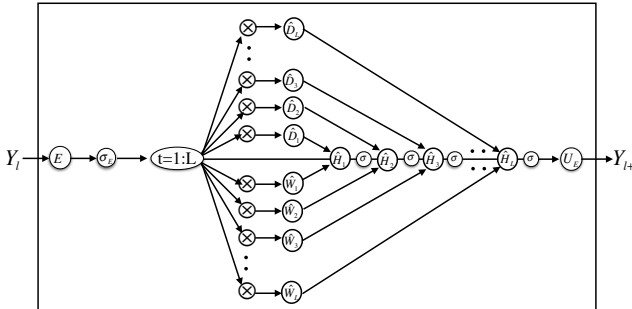

Figure 1: QUNN arcitecture: $\otimes$ represents tensor product of input with different time delays, oval marked by $t = 1 : L$ represents different timestep of delays, and $\sigma_*$ represents nonlinear activation functions at different layers.

|  | QUNN | LSTM | URNN | $n$-$t$-ORNN |
|---|---|---|---|---|
| memory scale | $n$ | 2 | 2 | $t$ |
| order of nonlinearity | $n$ | $n$ | $n$ | $n$ |
| stability | Yes | ? | Yes | ? |
| depth | $n + 2$ | $n$ | $n$ | $n$ |
| origin | Schrödinger equation | Ad Hoc | Unitarity | ODE |

Table 4: A top-down comparison between QUNN, LSTM, URNN and $n$-$t$-ORNN structure.

## 3.1 QUNN VS LSTM

With deeper understandings of RNN architectures provided in Sec. 1 in their nonlinearity and the memory scale of the underlying dynamics, we now analyze the connection between the quantum inspired RNN ansatz, proposed in the previous section, and one of the most widely used LSTM.

It is straightforward to see from the update rule of Eq. (15)–(15) that a QUNN with $L$ hidden layers corresponds to an $L^{th}$ order of nonlinearity in the ODE integration method. Moreover, notice that our matrix weight of the $n^{th}$ layers defined in Eq. (21) is determined by input state in the previous $n + 1$ step. Therefore, together QUNN with $L + 2$ total layers correspond to a $L$-$L$-ORNN. This means the functionality of QUNN is never equivalent to LSTM for any value of $L$. QUNN take time-sequential data as input to formulate weight matrix trainable through gradient based method to predict output at the same time step. LSTM however does not depend on the data directly.

We are now well-positioned to compare different RNN architectures from a top-down angle based on physical properties of their underlying dynamics according to different orders of nonlinearity and time-dependency as shown in Table. 4. More particularly, QUNN have longer range of memory scale than both LSTM and URNN, i.e., the order of time derivatives in the corresponding ODE is higher in QUNN. This comes with a price of additional layers of embedding in RNN architecture seen in the depth difference. But QUNN possess stability by construction, which is not guaranteed in generic LSTM architecture. This show cases the distinction between an ad hoc heuristic approach and physical inspired approach to designing RNN. In the problems where long-term dependencies are important, such RNN stability becomes essential to effective learning without been hampered by the exploding or vanishing gradients.

## 4 CONCLUSION

We propose a generalized framework of ODE neural networks with $n^{th}$ order linearity $t^{th}$ order time derivatives which includes many existing RNN architectures, including LSTM, GRU, URNN, CW-RNN, and the quantum inspried ansatz QUNN, as special cases. Our proof of the equivalence of between quantum dynamics and RNN propagation forms the basis of an RNN design, the QUNN. We show that this architecture is provably stable and possesses less complexity overhead in the dimension of input data than generic LSTM architectures.

Our findings support the concept of harnessing physical knowledge of the data for constructing the corresponding machine learning tools. Additional knowledge about the time correlation length that manifests memory scale of the dynamics and the degree of nonlinearity of the dynamics behind the input data will help the design of appropriate RNN suitable for the given problem.

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
