# OpenReview forum: "Unification of  Recurrent   Neural Network Architectures and Quantum Inspired Stable Design "
_ICLR.cc/2019/Conference_

### Official Review · AnonReviewer2 · 2018-11-02
**Interesting viewpoint, but could use more examples**

**Rating:** 5
**Confidence:** 2

**Review:**

In this paper, the authors relate the architectures of recurrent neural
networks with ODEs and defines a way to categorize the RNN architectures by
looking at non-linearity order and temporal memory scale. They further
propose QUNN, a RNN architecture that is more stable and has less complexity
overhead in terms of input dimension while comparing with LSTM.

Although this paper provides a new view point of RNN architectures and relates
RNNs with ODEs, it fails to provide useful insight using this view point.
Also, it is not clear what advantage the new proposed architecture QUNN has
over existing models like LSTM or GRU.

The paper is well presented and the categorization method is well defined.
However, how the order of non-linearity or the length of temporal memory
affect the behavior and performance of RNN architectures are not studied.

It is proved that QUNN is guaranteed existence and its Jacobian eigen values
will always have zero real part. It would be easier to understand if the
authors could construct a simple example of QUNN and conduct at least some
synthetic experiments.

In general I think this paper is interesting but could be extended in various
ways.

---

### Official Review · AnonReviewer1 · 2018-11-02
**Interesting ideas but unclear exposition**

**Rating:** 4
**Confidence:** 3

**Review:**

The authors make connections between RNN dynamics and those of a class of ODEs similar to RNNs (ORNN) that has different orders of nonlinearity and order of gradients in time. They show that typical RNN architectures can be described as members of the ORNN family. They then make the connection that quantum mechanical systems can be described as following the Schrodinger equation which can be cast as a series of coupled 1st order ODEs of the evolution of wavefunctions under an influencing Hamiltonian. They then claim that these discretized equations can be represented by a RNN similar to a unitary RNN. They go on to outline a RNN structure inspired by this insight that has time-dependent activations to increase the scale of temporal dependence.

The main challenge of this paper is that it does not present or support its arguments in a clear fashion, making it difficult to judge the merit of the claims. Given the nuance required for their arguments, a more robust Background section in the front that contextualizes the current work in terms of machine learning nomenclature and prior work could dramatically improve reader comprehension. Also, while the parallels to quantum mechanics are intriguing, given that the paper is arguing for their relevance to machine learning, using standard linear algebra notation would improve over the unnecessary obfuscation of Dirac notation for this audience. While I'm not an expert in quantum mechanics, I am somewhat proficient with it and very familiar with RNNs, and despite this, I found the arguments in this paper very hard to decipher. I don't think this is a necessity of the material, as the URNN paper (http://proceedings.mlr.press/v48/arjovsky16.pdf) describes very similar concepts with a much clearer presentation and background.

Further, despite claims of practical benefits of their proposed RNN structure, (reduced parameter counts required to achieve a given temporal correlation), no investigations or analyses (even basic ones) are performed to try and support the claim. For example, the proposed scheme requires a time varying weight matrix, which naively implemented would dramatically grow the parameter count over a standard LSTM. I can understand if the authors prefer to keep the paper strictly a theory paper, but even the main proof in Theorem 4 is not developed in detail and is simply stated with reference to the URNN paper.

There are some minor mistakes as well including a reference to a missing Appendix A in Theorem 3, "Update rule of Eq. (15)-(15)", "stble regime". Finally, as a nit, the claim of "Universal computing" in the name, while technically true like other neural networks asymptotically, does not seem particularly unique to the proposed RNN over others, and doesn't provide much information about the actual proposed network structure, vs. say "Quantum inspired Time-dependent RNN".

---

### Official Review · AnonReviewer5 · 2018-11-09
**Possibly interesting, but unclear exposition and lack of concrete evidence**

**Rating:** 4
**Confidence:** 2

**Review:**

This paper attempts to do three things:
	1) introduce a generalization / formalism for describing RNN architectures
	2) demonstrate how various popular RNN architectures fit within the proposed framework
	3) propose a new RNN architecture within the proposed framework that overcomes some limitations in LSTMs
The ultimate goal of this work is to develop an architecture that:
	1) is better able to model long-term dependencies
	2) is stable and efficient to train

Some strengths, concerns, and questions loosely ordered by section:

Stable RNNs
	- it's not clear to me where equations (2) and (3) come from; what is the motivation? Is it somehow derived from this Runge-Kutta method (I'm not familiar with it)?
	- I don't understand what this t^th order time-derivative amounts to in practice. A major claim (in Table 4) is that LSTMs are time-order 2 whereas QUNNs are time-order L and the implication is that this means LSTMs are worse at modeling long term structure than QUNNs ; but how does that actually relate to practical ability to model long-term dependencies? It certainly doesn't seem correct to me to say that LSTMs can only memorize sequences of length 2, so I don't know why we should care about this time-derivative order.
	- I thought this section was poorly written. The notation was poorly chosen at times, e.g. the t_k notation and the fact that some $l$ have subscripts and some don't. There were also some severe typos, e.g. I think Claim 1 should be "L-2-ORNN". Furthermore, there were crucially omitted definitions: what is reversibility and why should we care? Relatedly, the "proofs" are extremely hand-wavy and just cite unexplained methods with no further information.

QUNNs
	- The practical difference between QUNNs and LSTMs seems to be that the weights of the QUNN are dynamic within a single forward prop of the network, whereas LSTM weights are fixed given a single run (although the author does admit that the forget gates adds some element of dynamism, but there's no concrete evidence to draw conclusions about differences in practice).
	- I don't understand the repeated claim that LSTMs don't depend on the data; aren't the weights learned from data?

There may be something interesting in this paper, but it's not clear to me in its current incarnation and I'm not convinced that an eight-page conference paper is the right venue for such a work. There's a substantial of amount of exposition that needs to be there and is currently missing. I suspect the author knows this, but due to space constraints had to omit a lot of definitions and explanations.

I don't think all papers need experiments, but this paper I think would have greatly benefited from one. Community knowledge of LSTMs has reached a point where they are in practice easy to train and fairly stable (though admittedly with a lot of tricks). It would have been much more convincing to have simple examples where LSTMs fail due to instability and QUNNs succeed. Similarly, regarding long-term dependencies, my sense is that LSTMs are able to model some long-term dependencies. Experimental evidence of the gains offered by QUNNs would have also been very convincing.

Note: It looks like there's some funkiness in the tables on page 8 to fit into the page limit.

---

### Official Review · AnonReviewer4 · 2018-11-10
**Interesting idea, but difficult to read**

**Rating:** 5
**Confidence:** 2

**Review:**

This paper presents a new framework to describe and understand the dynamics of RNNs inspired by quantum physics. The authors also propose a novel RNN architecture derived by their analysis.

Although I found the idea quite interesting, my main concern is that the jargon used in the paper makes it hard to understand. I suggest that the authors to add an in-depth "background" section, so the reader becomes more familiar with the terms that will be introduced later.

Despite this paper is mainly a theory paper, it would have a lot more strength if the authors provide some experiments to demonstrate the strength of the proposed architecture over LSTMs.

As a minor suggestion, the term "universal" should be removed from "UNIVERSAL COMPUTING NEURAL NETWORK" as all recurrent neural networks are, in theory, universal.

---

### Meta-Review · Area_Chair1 · 2018-12-14
**rejection**

**Confidence:** 4
**Recommendation:** Reject

**Metareview:**

although the way in which the authors characterize existing rnn variants and how they derive a new type of rnn are interesting, the submission lacks justification (either empirical or theoretical) that supports whether and how the proposed rnn's behave in a "learning" setting different from the existing rnn variants.